# Mapping the Environmental Co-Benefits of Reducing Low-Value Care: A Scoping Review and Bibliometric Analysis

**DOI:** 10.3390/ijerph21070818

**Published:** 2024-06-22

**Authors:** Gillian Parker, Sarah Hunter, Karen Born, Fiona A. Miller

**Affiliations:** Collaborative Centre for Climate, Health & Sustainable Care, Institute of Health Policy, Management and Evaluation, University of Toronto, Toronto, ON M5T 3M6, Canada

**Keywords:** low-value care, environmental sustainability, healthcare, co-benefits, scoping review, bibliometric analysis

## Abstract

Reducing low-value care (LVC) and improving healthcare’s climate readiness are critical factors for improving the sustainability of health systems. Care practices that have been deemed low or no-value generate carbon emissions, waste and pollution without improving patient or population health. There is nascent, but growing, research and evaluation to inform practice change focused on the environmental co-benefits of reducing LVC. The objective of this study was to develop foundational knowledge of this field through a scoping review and bibliometric analysis. We searched four databases, Medline, Embase, Scopus and CINAHL, and followed established scoping review and bibliometric analysis methodology to collect and analyze the data. A total of 145 publications met the inclusion criteria and were published between 2013 and July 2023, with over 80% published since 2020. Empirical studies comprised 21%, while commentary or opinions comprised 51% of publications. The majority focused on healthcare generally (27%), laboratory testing (14%), and medications (14%). Empirical publications covered a broad range of environmental issues with general and practice-specific ‘Greenhouse gas (GHG) emissions’, ‘waste management’ and ‘resource use’ as most common topics. Reducing practice-specific ‘GHG emissions’ was the most commonly reported environmental outcome. The bibliometric analysis revealed nine international collaboration networks producing work on eight key healthcare areas. The nineteen ‘top’ authors were primarily from the US, Australia and Canada.

## 1. Introduction

Health systems are major contributors to climate change and other ecological harms, and are increasingly recognizing their obligation to mitigate their environmental impacts in the short term while moving toward health system sustainability in the long term [1,2,3]. The provision of healthcare contributes significantly to myriad environmental harms, including carbon emissions, pollution and waste. The most significant environmental impacts of healthcare come not from healthcare facilities themselves, but rather from the products and services that healthcare organizations deliver and consume [2,4]. Since the COP28 Summit, 150 countries have committed to putting health at the centre of climate action and flagging the need for climate resilient and low-carbon health systems [5]. Strategies identified in the literature as critical to ensuring an effective and efficient health system include reducing low-value care (LVC) [6,7] and reducing the environmental impact of healthcare [8,9,10]. Moreover, awareness is developing among clinicians, health system leaders and decision makers of the co-benefits that can be realized through interventions that integrate efforts to reduce the environmental impacts of care with efforts to reduce (LVC) [11,12,13].

Low-value care includes care practices (tests, treatments or procedures) that have been identified, using scientific evidence, to be unnecessary, ineffective or harmful in hospital, primary-care, long-term care or public-health contexts [14]. Common examples of LVC include antibiotics for viral infections and laboratory testing prior to low-risk surgeries [7]. Reducing LVC offers myriad benefits, including improving patient care and outcomes and freeing resources for expanded coverage [7,15,16,17]. Patients can be harmed by LVC, whether during the test or procedure or through infections or reactions, or by over-testing or over-treatment [7,17,18]. By definition, LVC generates carbon emissions, waste, and pollution without improving patient or population health [15,19]. In addition, LVC occurs most often in high-income country (HIC) settings—settings where access and resources are abundant [20] while the environmental impact of LVC, in the form of GHG emissions for example, disproportionally impacts marginalized groups and those in low- and middle-income country (LMIC) settings [12,21].

Within the climate change literature, “co-benefits” include the positive environmental impacts that a policy or intervention aimed at one objective might have on other objectives, thereby increasing the total benefit for society [22]. For health systems, addressing the challenge of LVC [11,13] has the potential to be a critically important strategy for securing environmental co-benefits at the frontline of care delivery and at organization and system levels. Approaches to reducing LVC that identify, address, measure and report on environmental co-benefits have the potential to produce “win–win” situations and to overcome silos to achieve objectives of improved health and reduced environmental impact. Co-benefits, within the climate and environmental-sustainability literature, have been described as “happy accidents” that produce a benefit. There may be potential to deliberately optimize such benefits in healthcare by understanding interdependent relationships, identifying synergies, and addressing potential barriers [23]. This approach was recently promoted by the National Institute for Health and Care Excellence (NICE)—the UK National Health Service organization that publishes health technology, clinical, and health promotion and social care guidelines—which acknowledged the environmental co-benefits of reducing LVC as a critical health-system strategy: “NICE can influence both direct and indirect carbon emissions, and already does so through its guidance and advice products and work to reduce the use of low value care” ([24], p. 4). With significant opportunity to benefit resource-constrained health systems, research- and practice-change interventions to advance the environmental co-benefits of reducing low-value care represent a valuable strategy for reducing both environmental and patient harms. While recognition and activity in this area of research and practice change is flourishing, to our knowledge, no reviews of the literature on the environmental co-benefits of reducing LVC have been published.

### Objectives and Research Questions

The objectives of this study were to identify and characterize a body of literature to build foundational knowledge and advance understanding of this field through a scoping review and bibliometric analysis. Specifically, a goal of this study was to illustrate the trends in the research and practice change literature, and especially to identify emerging areas of interest (focus) in this field. In addition, the study aimed to develop quantitative and visual data on the key authors, countries, networks and international trends advancing work at the intersection of environmental sustainability and reducing low-value care.

The research questions that motivated this study were the following:Scoping ReviewoHow has knowledge production in this field evolved over time?oWhat are the key and emerging areas of focus in both healthcare and environmental sustainability in this field?
Bibliometric AnalysisoWhat are the prolific collaborations in this field?oWho/what are the prolific authors, institutions, countries and publication sources in this field?


## 2. Materials and Methods

We selected a scoping review and bibliometric analysis as the ideal methods to conduct our inquiry. A scoping review, a literature-synthesis type, is most appropriate when examining emerging and/or broad topics with the aim of characterizing their features [25]. Bibliometric analysis involves descriptive, statistical analysis of aggregated bibliometric metadata associated with relevant publications to provide insights into the key topics and contributors (authors, author institutions and institution countries)—and the relationships between them—within a particular research area (field) [26,27,28,29]. We used the scoping review methodological framework proposed by Arksey and O’Malley [25] and later enhanced by Levac, Colquhoun, and O’Brien [30]. The Preferred Reporting Items for Systematic Reviews and Meta-Analysis Extension for Scoping Reviews (PRISMA-ScR) [31] also informed the conduct and reporting of the scoping review component (see Appendix A). Currently, no reporting checklist exists to support the conduct and/or reporting of bibliometric analysis. Accordingly, we developed and completed a reporting checklist, specific to bibliometric analysis, based on our knowledge of, and experiences with, the methodology (see Appendix A).

### 2.1. Search Strategy

A preliminary review of the literature was conducted to explore publications focused on the environmental co-benefits of reducing LVC and to inform the development of the search strategy. Based on this, we developed a 44-search-term strategy and conducted a comprehensive search of four database: Scopus, MEDLINE, and EMBASE, and CINAHL (see Appendix A). This search was first run in January 2023, capturing publications from each databases’ inception to January 2023, and then re-run in July 2023 to update the search to 1 July 2023. Search terms were categorized into three groups:*Low-value care (overdiagnos* or “low value” or low-value or overtest* or over-test* or overtreat* or over-treat* or “choosing wisely” or “less is more” or “reducing waste” or overuse or “minimal benefit*” or de-implement* or deimplement* or deadopt* or de-adopt* or de-prescrib* or unnecessary or “over surveillance”) AND;**Environmental sustainability (“environmentally sustainable” or “environmental sustainability” or “carbon footprint*” or “carbon emission*” or “climate change” or “net zero” or “climate risk” or “low carbon” or carbon* or de-carbon* or “carbon performance” or “indirect carbon impact*” or “environmental impact” or “GHG emission*” or “environmental emission*” or “greenhouse gas emission*” or carbon-constrained or “climate crisis”) AND;**Health (health* or healthcare or medicine* or medica* or “commissioned care” or hospital* or laborator*).*

### 2.2. Inclusion Criteria

Included publications focused (to varying degrees) on environmental co-benefits of reducing LVC. This focus could be primarily on reducing environmental harms of healthcare, with reducing LVC as a strategy, or primarily on reducing LVC with a recognition of the environmental co-benefits or (approximately) equally focused on both. These criteria were purposefully broad, as this is a nascent field and no literature was available regarding the scope of the field or possible data collection parameters. Included publications were not restricted by geography or healthcare setting. All types of publications were included (e.g., empirical studies, reviews, commentaries, editorials, conference abstracts) to capture all relevant knowledge production and comprehensively map its breadth and scope. Due to resource constraints, only English language publications were included.

### 2.3. Exclusion Criteria

Publications were excluded if they focused on non-human subjects, such as animals or agriculture, or were focused on natural or applied sciences (e.g., chemistry, earth science, engineering).

### 2.4. Publication Selection

Search results were imported into Covidence, a Cochrane technology platform (www.covidence.org), and duplicates were automatically removed. Title and abstract screening and full-text review of publications were conducted by three research team members (GP, SH, TB) who resolved discrepancies through discussion. Two researchers (GP and SH) screened all titles and abstracts independently. One research team member (GP) reviewed a random 10% sample of screened abstracts and resolved discrepancies. A full-text review was conducted by two researchers (GP and SH), and discrepancies were discussed and resolved collaboratively. The reference lists of included publications were hand searched to identify additional, relevant publications.

### 2.5. Scoping Review—Specific Methods

#### 2.5.1. Data Collection

The scoping review data-collection worksheet was designed iteratively; it was piloted with 10 included publications and revised accordingly. Data collected included publication characteristics (publication year, type of publication), healthcare focus and environmental sustainability focus.

#### 2.5.2. Data Analysis

Data were entered into an Excel-based spreadsheet to facilitate data analysis and reporting. Data analysis was conducted by two research team members. Descriptive statistics were used to summarize the data. The data were analyzed by three members of the research team (GP, SH and FM) with discrepancies resolved collaboratively. All members of the research team reviewed the final summary of results.

#### 2.5.3. LVC and/or Environmental Sustainability Focus

While all publications included the environmental co-benefits of reducing LVC, the degree and amount of focus varied significantly. For example, some publications focused primarily on reducing the environmental impacts of healthcare and listed reducing LVC as a strategy, while others focused on reducing LVC with a recognition of the environmental co-benefits; finally, a subset focused on both aspects equally, reporting both reducing-LVC and environmental-sustainability outcomes. To document this characteristic, we categorized the publications in three groups:Publications (approximately) equally focused on environmental sustainability of healthcare AND reducing LVC;Publications primarily focused on environmental sustainability of healthcare (and acknowledging the benefits of reducing LVC); Publications primarily focused on reducing LVC (and acknowledging the environmental co-benefits of doing so).

### 2.6. Healthcare Focus

Categorizing the healthcare focus of these publications was complex, as publications crossed disciplines, medical practices and healthcare settings. An inductive process was used by one researcher (GP) to map the categories and sub-categories, then validated by two research team members (SH and FM). This process was iterative, included numerous discussions, and was data led. The categories used for this analysis reflect a nascent dataset, and offer the opportunity to be refined and standardized as work in this field continues. To this end, our categorization demonstrates the breadth and scope of healthcare focus in this field, but is not definitive nor necessarily exhaustive. We developed categories across four areas of healthcare focus (with 15 sub-categories): ‘Procedures’ (laboratory testing, imaging, respiratory, anesthesia, surgery, Intensive Care Unit (ICU)), ‘System organization/design/evaluation’ (general healthcare, measurement/metrics), ‘Pharmaceuticals’ (antibiotics and other) and ‘Care Type/Setting’ (hospital, primary, mental health psychiatry, nursing, other).

### 2.7. Environmental Focus

As this is an emerging field, we decided to only collect data related to environmental sustainability for the subset of empirical papers we categorized as (approximately) equally focused on environmental sustainability of healthcare and reducing LVC. Our rationale was that we wanted to understand which aspects of environmental sustainability were being studied empirically when both environmental sustainability and reducing LVC were studied as a common focus (rather than parenthetically attending to one or other outcome).

Categorizing the environmental sustainability focus was also complex, as publications often used vague or differing definitions of relevant terms and focused on broad areas of environmental sustainability. An inductive process was used by one researcher (GP) to map the categories and sub-categories, and then validated by two research team members (SH and FM). Similar to the healthcare focus, this process was iterative, included numerous discussions, and was data led. We developed categories across six areas of environmental sustainability (with sixteen sub-categories): ‘GHG emissions’ (resulting from both general healthcare and specific practices), ‘Pollution’ (air, land, water), ‘Resource use’ (use fewer single-use products, water, energy/electricity, natural /non-renewable/ raw resources), ‘Waste management’ (recycling, compostability, re-use, re-process, produce less solid waste, produce less waste (general), ‘Supply chain and facility/service design’ (service design/ processes, facilities/building/ system, sustainable supply chain, sustainable procurement), and ‘Environmental stewardship’ (education (decision makers, providers, patients, public) and support/ influence/actions of staff, suppliers, etc.).

As work on extending the breadth and scope of environmental sustainability addressed in healthcare is growing in prominence, our goal was to collect data on both primary (outcomes) and secondary (evidence cited or recommendations made) information from these empirical studies. We developed two groupings for this information and, within each of the sub-categories, we collected data and classified them as either: (i) evidence or recommendation, if a study cited existing evidence or made a recommendation in an environmental sustainability category; or (ii) reported outcomes, if a study reported new study outcomes in an environmental sustainability category.

### 2.8. Bibliometric Analysis-Specific Methods

#### 2.8.1. Data Collection

Raw metadata were retrieved for all publications analyzed in the scoping review, primarily from Web of Science (WoS) (as per bibliometric analysis guidance) [27] with additional data retrieved from Embase, Scopus, and Medline, as necessary.

#### 2.8.2. Data Cleaning

The raw metadata were converted in preparation for data cleaning, analysis, and visualization using Biblioshiny [27]. Data cleaning ensured the quality of data inputs, and, in turn, data outputs. Data cleaning focused on metadata related to author names (WoS field tag ’AU’), author institutions (the affiliate/s the author (and their work) is associated with at the time of publishing; ‘C1′), and publication sources (’SO’). Specific actions taken to clean the metadata included the following: removing duplicate items; retrieving and inputting ‘missing’ data from alternative sources; ensuring consistent formatting of data; and ensuring no data entry errors (e.g., misspellings, improper punctuation, etc.) (see Appendix A). Data cleaning was facilitated using Microsoft Excel v. 16.86 and OpenRefine (https://openrefine.org/) software applications.

#### 2.8.3. Data Analysis and Visualization

Two analytic techniques were employed: publication [28], and co-authorship analysis [26]. Microsoft Excel, OpenRefine, and Biblioshiny [27] software applications were used to facilitate data analysis and visualization. Microsoft Excel- and Biblioshiny-generated data visualizations were recreated, and additional layers of analysis were applied as necessary, using the vector-based graphics software, Adobe Illustrator (see Appendix A).

## 3. Results

### 3.1. Literature Search

The database searches identified 2456 publications (after duplicates were removed), for which the titles and abstracts were screened for inclusion. Of these, 372 publications were selected for full-text screening and 145 publications were included in these analyses (see Appendix A). Figure 1: a flow diagram we developed, inspired by the PRISMA flow diagram, which describes the stepwise process used to identify relevant publications to be analyzed in both the scoping review and bibliometric analysis components of this study (n = 145). Our enhanced diagram for each component of this study describes how the data were analyzed and synthesized (e.g., identification of analysis-type, software used, etc.), visualized (e.g., identification of software used, whether post-work visualization was completed, etc.,) and, ultimately, reported in this manuscript (e.g., identification of reporting format of each analysis).

### 3.2. Scoping Review—Specific Results

#### 3.2.1. Publication Timeline

The first included paper was published in 2013. The majority of included publications (34%) were published within the first half of 2023, with only 12% of publications produced before 2020.

#### 3.2.2. Type of Publication

The most prominent publication types were commentaries/opinions/editorials/viewpoints (51%), followed by reviews (23%) and empirical studies (21%). The remaining 5% of publications included protocols, conference abstracts and position statements.

#### 3.2.3. LVC and/or Environmental-Sustainability Focus

The majority of the publications were primarily focused on environmental sustainability of healthcare, with reducing LVC as a suggested strategy (52%), followed by publications (approximately) equally focused on both environmental sustainability and reducing LVC (34%), followed by publications primarily focused on reducing LVC with a recognition of the environmental co-benefits (14%).

#### 3.2.4. Healthcare Focus of Publications

Healthcare focus was recorded in four categories—‘Procedures’, ‘System organization/design/evaluation’, ‘Pharmaceuticals’ and ‘Care Type/Setting’ and 15 sub-categories (Figure 2). The first category, ‘Procedures’, captured the majority of publications (42%) and included procedures related to the following subcategories: laboratory testing, imaging, respiratory, anesthesia, surgery and ICU. McAlister et al. [32], in their study of the carbon footprint of pathology testing, note that “Pathology testing is often not clinically indicated and unneeded testing can produce higher numbers of false positive results, leading to further unnecessary testing, overdiagnosis of disease, and unnecessary and potentially harmful treatment. Each element in this cascade also has an associated detrimental environmental impact” (p: 381). The second category, ‘System organization/ design/ evaluation’, captured publications (30%) focused on the health system or healthcare generally or publications related to metrics or measurement. For example, in their commentary, High value health care is low carbon health care, Barratt and colleagues [11] advocate for broad health-system changes: “This is where two key challenges to health system sustainability—low value care and climate risk—intersect, and why better value, low carbon emissions models of clinical care are urgently needed” (p: 67). The third category, ‘Pharmaceuticals’, (14%), was split into antibiotics and other pharmaceuticals to demonstrate the amount of work published specific to antibiotics. The fourth category, ‘Care Type/Setting’, captured publications (14%) focused on care within a particular setting rather than a specific procedure or pharmaceutical. This category included primary care-, hospital-, mental health/psychiatry- and nursing-focused publications.

#### 3.2.5. Environmental Focus of Publications

As discussed in the Methods section, we collected data related to environmental sustainability for the subset of empirical papers we categorized as (approximately) equally focused on environmental sustainability of healthcare and reducing LVC. For the 13 empirical studies included in this analysis, we reported results across six categories, ‘GHG emissions’, ‘Pollution’, ‘Resource use’, ‘Waste management’, ‘Supply chain and facility/service design’ and ‘Environmental stewardship’ and sixteen sub-categories of environmental outcome (Figure 3 below).

As described in the Methods section, within each of the sub-categories, data were classified as (i) evidence or recommendation, if a study cited existing evidence or made a recommendation from an environmental sustainability category; or (ii) reported outcomes, if a study reported new study outcomes in an environmental sustainability category. Evidence or recommendation data were reported across all 16 sub-categories. The majority were for ‘GHG emissions (healthcare general)’, ‘GHG emissions (specific healthcare practice)’ and ‘Pollution’. Reported outcomes were present in 11 of the 16 subcategories. Of the thirteen studies that reported outcomes, seven reported outcomes across multiple categories and four reported a single outcome. Eleven of the thirteen studies reported a reduction in ‘GHG emissions’, followed by outcomes for ‘Use less single-use products’, then outcomes for ‘Use less energy’. For example, in their empirical study on reducing unnecessary bloodwork, Spoyalo and colleagues [33] provided a context, noting that the GHG emissions they were reporting on were associated with the full life-cycle—the production, transport, processing and disposal of the consumables associated with laboratory testing—but excluded processes such as heating, ventilation, air conditioning and refrigeration, as their energy consumption does not vary with laboratory testing volumes.

### 3.3. Bibliometric Analysis-Specific Results

#### 3.3.1. Top Author Production over Time

Five hundred and eighty-one unique authors contributed to the 145 included publications between 2013 and 2023 (July). There were 19 ‘top’ authors (authors with three or more publications), 14 of whom had already published relevant work in 2023. Most authors began publishing relevant publications from 2020 onwards. Seven of the ‘top’ authors were from the USA; six from Australia; three from Canada; two from the UK; and one from The Netherlands.

#### 3.3.2. Author Collaboration Networks

Figure 4 depicts the top 10% of authors who have collaborated on at least one publication, and describes the networked relationships between these authors. One node (the grey-coloured circle) represents one author; a solid grey line represents at least one collaboration between a pair of authors within a network (cluster of nodes); and a dashed grey line represents at least one collaboration between a pair of authors across networks. The closer the nodes, the stronger the collaborative relationship. Nine distinct networks were identified. The largest networks, by number of collaborators involved (13 each), were centered around Forbes McGain in Australia (whose included publications were published between 2019 and 2023, and who collaborated with all authors in the network) and Jodi Sherman in the USA (whose included publications were published between 2020 and 2023, and who collaborated with all but one author in the network).

#### 3.3.3. Country Collaboration Networks

Figure 5 depicts the country collaboration networks based on author institutions. One node (a coloured circle) represents one country. The closer the nodes, the stronger the collaborative relationship. Two primary networks were identified (blue and red). The largest network by number of countries captured (blue) comprised 17 unique countries, and represented seven publications published between 2019 and 2023 and generated by two or more of the authors within the network (but not exclusively). The second largest, but most productive, network (red) comprised 11 unique countries, and represented 24 publications published between 2019 and 2023 and generated by two or more of the authors within the network (but not exclusively). The remaining networks (purple, green, orange, pink, brown, and grey) each represent one publication (n = 6) published between 2019 and 2023 and generated through collaboration with authors situated within the primary networks (blue and red).

#### 3.3.4. Top Institutions

Three hundred and eighty-seven unique institutions (affiliates) contributed to the 145 included publications. Figure 6 depicts the top institutions (with four or more publications). There were sixteen ‘top’ institutions: five were from Australia; four from the USA; four from the UK; and three from Canada. The top-producing institution was the University of Sydney, with 18 publications. The majority of top-producing institutions were universities, followed by medical centres or medical organizations.

#### 3.3.5. Top Journals

One hundred and seven unique journals contributed to the 145 included publications. There were six ‘top’ journals (journals with three or more publications). *The BMJ* was the top-producing journal, with twelve publications, followed by *Healthcare Papers* and *The Journal of Climate Change and Health* with four each, and *The Medical Journal of Australia*, *Resources Conservation and Recycling* and *Social Science and Medicine* with three each.

## 4. Discussion

The results of this study provide important insights into the emerging literature on the environmental co-benefits of reducing LVC. This section offers a detailed discussion of the key findings, applications, and directions for future work.

The publication trend over time demonstrated that research and evaluation to inform practice changes in this area has dramatically increased over the last three years. The fact that there were more publications in the first half of 2023 than in all of 2022, and that the majority of the top authors have published in the first half of 2023, shows significant momentum within the field. While the majority of publications to date are commentaries/perspectives/opinions, this characteristic is typical of an emerging field. The field will benefit from more empirical studies to assess the environmental harms resulting from healthcare and how these harms are reduced or eliminated when LVC is reduced or eliminated. Future empirical work should endeavour to conduct rigorous investigations and measure and report on the broad spectrum of environmental harms.

The collaboration network analyses revealed that large, international groups of authors are working together to advance this field. The largest author collaboration networks are centered around Australian and American authors, and the country collaboration-network analysis revealed that the most productive networks were spearheaded by authors in the UK, USA, Canada and Australia. The two largest author networks have published on multiple areas of healthcare (e.g., each of these networks has publications about the environmental co-benefits of reducing low-value care in general healthcare, lab testing imaging and infometrics/metrics/ measurement, and anesthesia). These results demonstrate that these author collaboration networks are not necessarily focused on a specific area of healthcare, but rather focused on co-benefits research and/or practice change. In addition, the results show that in recent years, increasingly, a greater number of authors from countries with emerging economies (such as Thailand, India, Jamaica, and Sudan) have been contributing to knowledge production in this field. While the community carrying out work in this field is small and quite concentrated in a few countries and institutions, the volume and momentum behind this work provides a significant opportunity for knowledge sharing and consensus development.

Our findings also demonstrated that the majority of included publications focused on environmental sustainability while flagging the importance of reducing LVC as a possible strategy. Importantly, the number of publications categorized as ‘(approximately) equally’ focused on the environmental co-benefits of reducing LVC has increased steadily over time, and these publications arguably best advance the agenda of realizing the environmental co-benefits of reducing LVC. Scholarship is beginning to present the importance of reducing LVC and environmental harms as inextricably linked. Thiel and Ritchie [12] describe the pernicious cycle of “harm, treat, harm” to describe the paradox of healthcare harming the environment, which in turn harms human health and requires more healthcare, which further harms the environment. For example, air pollution is known to induce breathing difficulties, and inhalers are used to minimize the effects of air pollution, but inhaler use generates a significant amount of carbon dioxide, which then exacerbates air pollution [12]. This cycle makes reducing LVC even more poignant, as care that produces environmental harms—but no benefits to patients—represents the antithesis of healthcare’s mandate.

The healthcare focus results highlighted the fact that the included publications covered a broad scope and diverse practices in healthcare. To capture this diversity, we developed 15 ‘healthcare focus’ sub-categories as a framework to begin to organize this work across system organization/design/evaluation, procedures, pharmaceuticals and care type/settings. While the scope was broad, the majority of publications were categorized as focused on general healthcare. The included empirical publications focused on targeted practice-change interventions for specific healthcare practices, primarily reducing unnecessary laboratory testing and inappropriate inhaler use. These two areas are also significant foci for research strictly focused on reducing environmental harms or reducing LVC, and represent a logical merging of these fields.

The analysis of the environmental sustainability focus demonstrated that the included empirical studies cited evidence, made recommendations, and reported outcomes across a broad spectrum of environmental sustainability outcomes. While we recognize the sample size was small and the categorization framework broad, our goal was to map essential aspects of environmental harm and efforts, through reducing LVC, to reduce it. The strong focus on GHG emissions in the empirical studies is likely due to the fact that GHG emissions are the most available environmental data in healthcare and can be translated into relatable results (e.g., equivalency to driving distances). While valuable, this focus also highlights the need to develop ways to evaluate all environmental sustainability impacts, collect rigorous data, and monitor and report on broader outcomes. In addition to GHG emissions, studies focused on solid waste management and recycling, which are established areas of environmental improvement. Of note, the included studies also extended into important, emerging areas of environmental sustainability such as composability, reprocessing or reuse [19,34,35]. Sustainable supply chain and procurement [33,34] and environmental stewardship [34,35,36] are also critical environmental issues beginning to receive attention in this literature. We acknowledge that the outcomes we have categorized (using fewer resources, better waste management, and more sustainable procurement) will all result in fewer GHG emissions, but it is important to explicate these processes as healthcare providers and organizations need to develop specific interventions to address these environmental issues.

Reducing LVC is a critical strategy to bolster health system sustainability, and the recognition of the environmental co-benefits of reducing LVC is gaining prominence and momentum. Thiel and Richie [12] describe ‘rejecting health care overuse’ as an act of beneficence by reducing both patients and global citizens’ risk of climate change-induced health harms. As mentioned, the majority of LVC occurs in HIC settings, while environmental harms disproportionally impact LMIC settings [20,21]. It is also important to note emerging research that identifies increasing overuse issues in low-resource settings [37]. The pervasiveness of LVC further supports global research and evaluation to inform practice change in this area, as overuse in LMIC settings is further exacerbated by limited public budgets, access to resources, and complex population health needs [36].

### Limitations

This study has several limitations. The search strategy was complicated by numerous factors such as a lack of agreed-upon search terms or MeSH terms, and the inability to search on terms like ‘environment’, ‘sustainable’, waste’, inappropriate’ (as these terms have broad meaning in healthcare and pull millions of results). To mitigate these issues, we iteratively developed the search strategy. While we acknowledge that our final set of ‘environmental’ terms appears to be very carbon-focused, we feel confident that our search was exhaustive. Publications in languages other than English were not included due to a lack of resources. In addition, while there is no clear consensus in the bibliometric analysis guidance literature regarding the generalizability of bibliometric study outputs to the field under study, we believe the systematic, comprehensive, and detailed approach used to identify relevant publications provides a robust understanding of the “co-benefits” field, despite not necessarily representing a “complete” population. Finally, the data collection categories were developed iteratively, based on the results within the included publications. Our goal is that these categorizations can be refined and standardized as work in this field continues to mature.

## 5. Conclusions

This foundational study is an important first step to identify research and evaluation to inform practice change with regards to the environmental co-benefits of reducing LVC and the authors and institutions carrying out this important work. The exponential growth in publications demonstrates a growing field of international collaboration and broad engagement across healthcare and environmental-sustainability outcomes. While the environmental focus of the field was predominately carbon-focused, the included publications addressed emerging areas such as composability, reprocessing or reuse, sustainable supply chain and procurement, and environmental stewardship. This review also highlights a need for empirical studies to advance practice change in this area. By systematically and comprehensively collecting and analyzing data on this emerging field, our research supports evidence-based health-system improvement work with the potential to increase effectiveness and efficiencies in resource-constrained health systems. Future research should focus on conducting rigorous empirical studies in this area, including the evaluation of and reporting on the broad spectrum of environmental harms.

## Figures and Tables

**Figure 1 ijerph-21-00818-f001:**
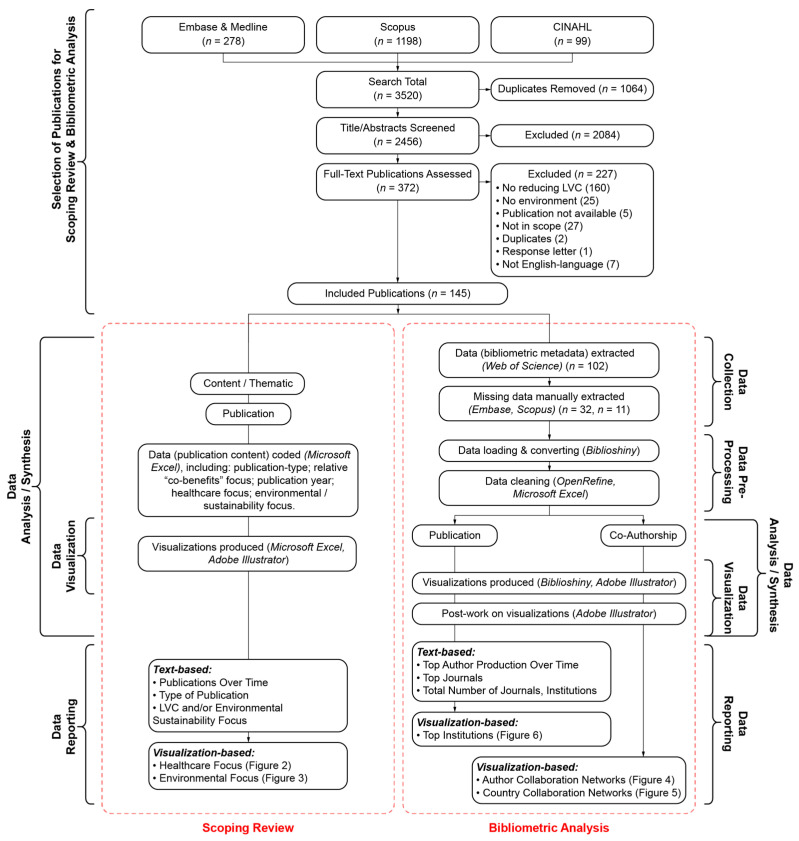
Flow diagram outlining the study design.

**Figure 2 ijerph-21-00818-f002:**
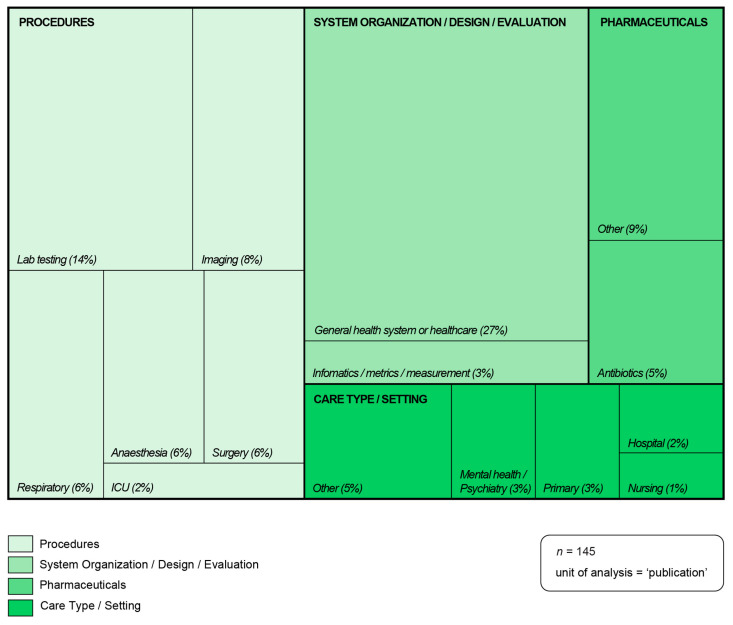
Healthcare focus of publications. Source: Microsoft Excel: analysis; Adobe Illustrator: visualization.

**Figure 3 ijerph-21-00818-f003:**
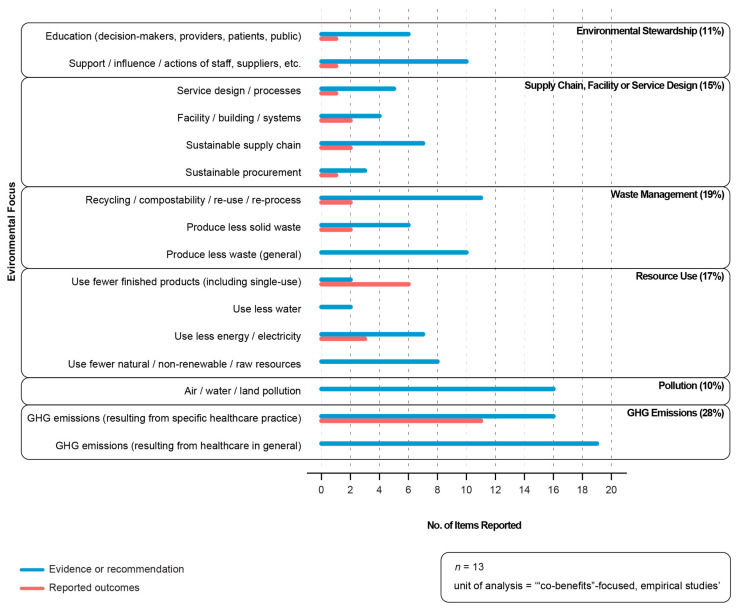
Environmental focus of publications. Source: Microsoft Excel: analysis; Adobe Illustrator: visualization.

**Figure 4 ijerph-21-00818-f004:**
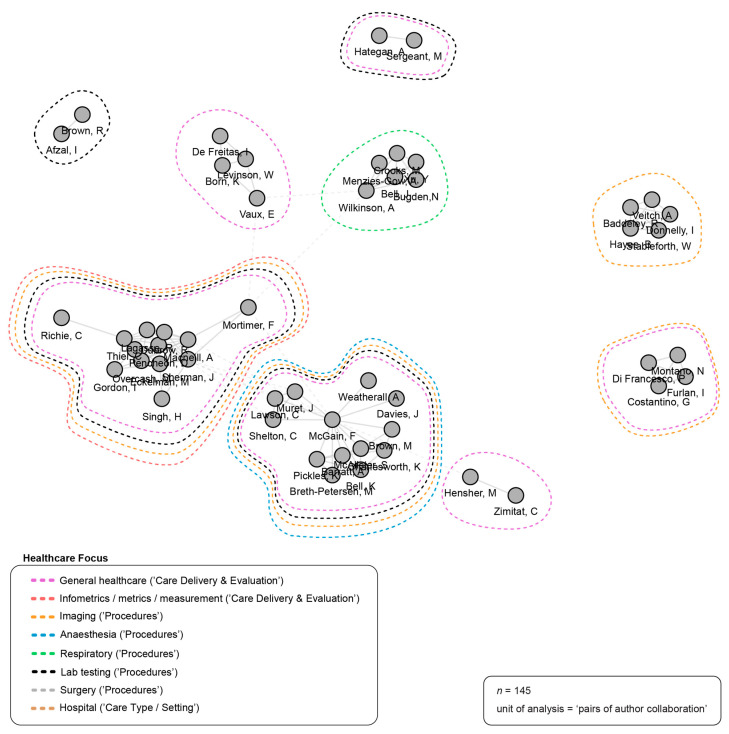
Author collaboration networks. Source: R-based application Biblioshiny was used; additional analysis (i.e., healthcare focus of publications) was completed by the study authors and applied to the visualization using Adobe Illustrator. Parameters specified: analysis: collaboration network; field: authors; network layout: automatic (default); clustering algorithm: walktrap (default); normalization: association (default); number of nodes: 58 (top 10% of total authors); repulsion force: 0.1 (default); remove isolated nodes: yes (default); minimum number of edges: 1 (default).

**Figure 5 ijerph-21-00818-f005:**
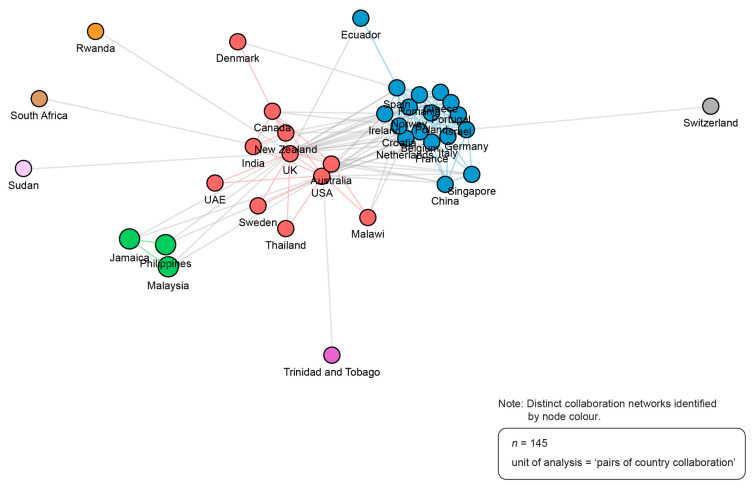
Country collaboration networks. Source: R-based application Biblioshiny was used; additional analysis (i.e., number of publications generated by each network) was completed by the study authors; visualization was recreated using Adobe Illustrator. Parameters specified: analysis: collaboration network; field: countries; network layout: automatic (default); clustering algorithm: walktrap (default); normalization: association (default); number of nodes: 200 (all countries in world, rounded); repulsion force: 0.1 (default); remove isolated nodes: yes (default); minimum number of edges: 1 (default).

**Figure 6 ijerph-21-00818-f006:**
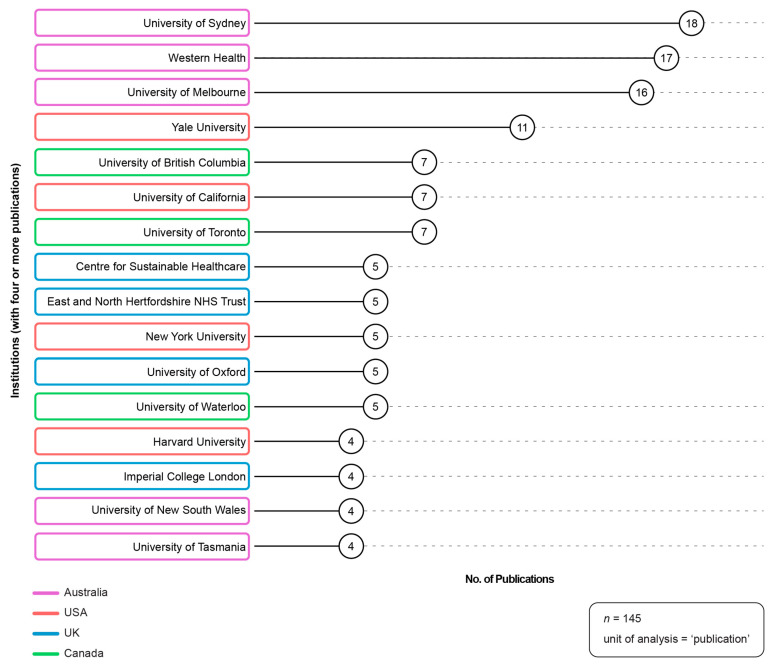
Top institutions. Source: R-based application Biblioshiny was used; additional analysis (i.e., institution country of origin; institution type) was completed by the study authors and applied, as necessary, to the visualization using Adobe Illustrator. Parameters specified: analysis: most relevant affiliations; affiliation name disambiguation: no; number of affiliations: 16.

## Data Availability

Data are provided in Appendix A.

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
