# Peer review of "Mapping the Environmental Co-Benefits of Reducing Low-Value Care: A Scoping Review and Bibliometric Analysis"

_ijerph, 2024, doi:10.3390/ijerph21070818_

Round 1

Reviewer 1 Report

Comments and Suggestions for Authors

Dear authors, thank you for your contribution. It is interesting and valuable. Below are my comments.

Abstract.

The findings of the study should be more detailed.

Introduction

I would recommend presenting some data on the negative impact of the health system on climate change and other environmental damage. In addition, after a brief presentation of the broad impacts, you should justify the focus of your analysis. Furher, you should provide a definition of sustainability and a common operationalisation of this concept.

Line 78-80: The aim of your study is too general. I would recommend that you refine it.

Line 81-85: Please critically review the research questions and distinguish between the bibliometric analyses and the scoping review.

Methods/Results:

Line 175-177/Line 192-195: Please give a more detailed description of how the categories have been developed.

Line 222-223: Please describe in more detail the data quality and the data cleaning process.

Line 241: I appreciate the idea of providing a holistic figure that covers all the steps of the analysis. Unfortunately, at the moment - especially without any explanation - the figure is confusing.

Line 249: Think about whether or not figure 2 is really necessary.

Line 290 (for example): It would be interesting to include some examples from the publication, e.g. what kind of recommendations were made?

Bibliometric analysis: I miss the bibliometric analysis of the content of publications.

Discussion: I think your results allow you to be more detailed in your discussion. At the moment it is too descriptive and not enough analytical. For example, it would be interesting if you could observe thematic differences over time and how they relate to the discourse on sustainability in general and the healthcare system in particular. In this context (which is also lacking in your work), it would be interesting to find out what concept of sustainability underlies the publications, as this has enormous consequences for the analytical approach in the selected articles. It would also be interesting to find out whether there are differences between authors and countries in terms of the topics studied. This would also make it possible to combine the results of the scoping review and the bibliometric analysis. These aspects are only examples.

Limitations: I would recommend providing more details. You should also discuss the categorisation of your findings.

Conclusion: Currently too general, please revise.

Reviewer 2 Report

Comments and Suggestions for Authors

Thank you for undertaking this very valuable and useful review of the literature and collaborations on this important and emerging topic. 

In my view, the methods adopted are suitable, the findings plausible, the implications for future work well-considered, and the paper itself well-written. The supplementary information also adds value.

I only have two very minor comments you might wish to consider:

Lines 34-36: 149 countries have now signed the COP28 Declaration on Climate and Health https://www.cop28.com/en/cop28-uae-declaration-on-climate-and-health

Line 39: update the reference to the 2023 Lancet countdown report, unless the point is made more strongly in the 2022 version.

Author Response

Reviewer Comment

Action

Lines 34-36: 149 countries have now signed the COP28 Declaration on Climate and Health https://www.cop28.com/en/cop28-uae-declaration-on-climate- and-health

We have updated the reference.

Line 39: update the reference to the 2023 Lancet countdown report, unless the point is made more strongly in the 2022 version.

We have updated the reference.

Round 2

Reviewer 1 Report

Comments and Suggestions for Authors

 Dear authors,

Thank you for the thorough revision. I still think it would have been valuable to add a thematic analysis with the bibliometric approach. But I also see that your focus in this analytical part of the study was on the research field itself.